# Site-selective C-H hydroxylation of pentacyclic triterpenoids directed by transient chiral pyridine-imino groups

Tong Mu[1], Bingcheng Wei[1], Dapeng Zhu[1] & Biao Yu [1,2✉]

Pentacyclic triterpenoids (PTs) constitute one of the biggest families of natural products, many with higher oxidation state at the D/E rings possess a wide spectrum of biological activties but are poorly accessible. Here we report a site-selective C-H hydroxylation at the D/E rings of PTs paving a way toward these important natural products. We find that Schönecker and Baran's Cu-mediated aerobic oxidation can be applied and become site-selective on PT skeletons, as being effected unexpectedly by the chirality of the transient pyridine-imino directing groups. To prove the applicability, starting from the most abundant triterpenoid feedstock oleanane, three representative saponins bearing hydroxyl groups at C16 or C22 are expeditiously synthesized, and barringtogenol C which bears hydroxyl groups at C16, C21, and C22 is synthesized via a sequential hydroxylation as the key steps.

[1] State Key Laboratory of Bioorganic and Natural Products Chemistry, Center for Excellence in Molecular Synthesis, Shanghai Institute of Organic Chemistry, University of Chinese Academy of Sciences, Chinese Academy of Sciences, Shanghai 200032, China. [2] School of Chemistry and Materials Science, Hangzhou Institute for Advanced Study, University of Chinese Academy of Sciences, 1 Sub-lane Xiangshan, Hangzhou 310024, China. ✉email: byu@sioc.ac.cn

**P**entacyclic triterpenoids (PTs) constitute a vast family of natural products, which have attracted great attention because of their significant biological and pharmacological activities, such as the antitumor, anti-inflammatory, and antiviral activities[1–4]. However, in-depth studies on the drugability of these complex molecules have been hampered by their poor accessibility. PTs occur usually in a heterogeneous manner in nature, with the structural diversity being greatly augmented by post-modifications on the basic pentacyclic skeletons, including especially the enzymatic C–H bond oxidation and *O*-glycosylation[5,6]. With the emergence of new C-H bond oxidation methodologies[7–11], a new stage is to set for chemical diversification of the naturally abundant PT feedstocks[12,13], such as oleanolic acid, which are always at low oxidation states.

Major types of the PTs include oleananes, ursanes, lupanes, and friedelanes, in that the hydroxyl groups are commonly present at C2, C3, C12, C16, C21, C22, and C23 (Fig. 1). Exploiting the biogenetic 3-OH, Baldwin et al. pioneered the directed C-H oxidation on PTs to introduce the 23-OH via a C3-oxime-mediated cyclopalladation procedure[14–16]. Baran et al. systematically explored C-H oxidation on the lupane skeleton; iodination could take place at C12 directed by a hydroxyl radical at C20 or C28, and remarkably, non-directed oxidation at C16 was realized with methyl(trifluoromethyl) dioxirane[17]. We have tried utilizing the innate C28-carboxylic acid group in oleanolic acid as a handle to derivatize the proximal CH₂ units; a palladium promoted dehydrogenation at C15–C16 or C21–C22 was achieved with 8-aminoquinoline amide or 2-aminomethylpyridine amide as the directing group[18]. Unfortunately, subsequent removal/transformation of the C28-amide auxiliaries was not successful. Indeed, practically useful oxidation/functionalization of the CH₂

units in the D/E rings of PTs remains a challenge[19]. Herein, we report an effective method to address this problem and its application to the successful synthesis of four structurally representative and biologically active high-oxidation-state PTs, these include PT glycosides (1–3)[20–24] bearing hydroxyl group at C16 or C22 and barringtogenol C (4)[25,26] bearing hydroxyl groups at C16, C21, and C22.

## Results

**Site-selective C–H hydroxylation of PTs.** Starting with oleanolic acid, the most abundant natural PT, we prepared a series of derivatives to explore the C–H oxidations directed by the C28 functional residues (Supplementary Fig. 1). Various protocols have been tried, including those developed by the research groups of Sanford[15], White[27], Yu[28,29], Dong[30,31], Suarez[32,33], and Baran[19], but met little success (Supplementary Fig. 2 and Supplementary Table 1). To our delight, applying Schönecker-Baran's Cu-mediated aerobic C–H hydroxylation[34–39], with pyridin-2-ylmethanamine (D1) as a transient directing group, to C28-aldehyde 5 led to 16β-ol derivative 5-a in 51% yield (Fig. 2, entry 1a). The instable nature of the imine intermediate might account for the moderate yield, we thus decided to replace primary amine D1 with a bulkier secondary amine[40], i.e., (pyridin-2-yl) ethan-1-amine D2, as the directing group.

Surprisingly, when racemic D2 was used in the reaction, the yield of 5-a dropped to 40%, while 22α-ol derivative 5-b was newly isolated in 20% yield (entry 1b). The structures of these isomers were determined unambiguously by X-ray diffraction analysis on their derivatives (S20 and S21, Supplementary Fig. 3). This outcome implied that the chiral methyl group could influence the orientation of the corresponding imine-pyridine-complexed copper intermediate toward either the 16β-H or 22α-H, which are both at the proximal γ position to the imino nitrogen[34–39]. Indeed, when enantiopure D2(**R**) was used, the yield of 5-b was increased to 40%, whereas the yield of 5-a decreased to 25% (entry 1c). In contrast, the use of D2(**S**) gave rise to 5-a exclusively in 70% yield (entry 1d). With the larger D3

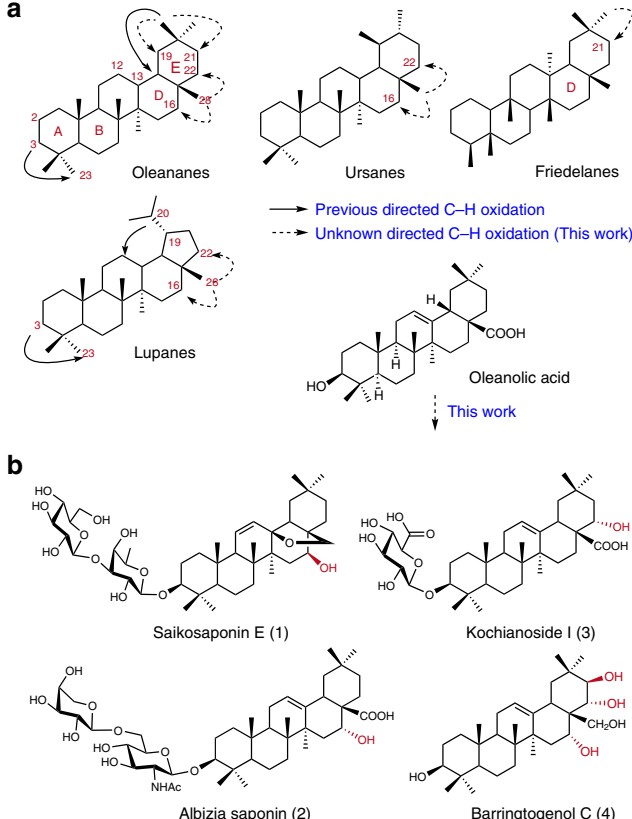

**Fig. 1 Overview of the present work. a** Typical PT skeletons and the common oxidation sites. **b** Synthetic targets of PT glycosides (**1–3**) and barringtogenol C (**4**) from oleanolic acid.

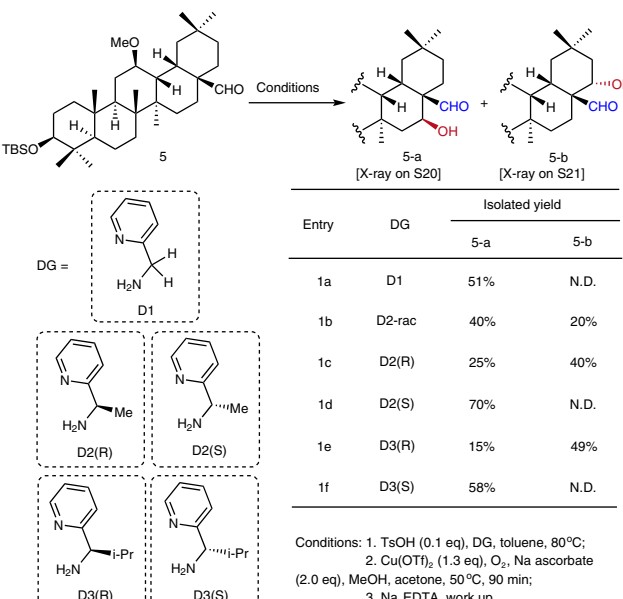

**Fig. 2 Site-selective C–H hydroxylation of oleanane C28-aldehyde 5 directed by transient (pyridin-2-ylmethyl)imino groups.** *i*-Pr isopropyl, Me methyl, Na₄EDTA ethylenediaminetetraacetic acid tetrasodium salt, N.D. not detected. TBS *tert*-butyldimethylsilyl, Tf trifluoromethanesulfonyl, Ts *p*-toluenesulfonyl.

|       |        | Isolated yield | |
|-------|--------|-----|-----|
| Entry | DG     | 5-a | 5-b |
| 1a    | D1     | 51% | N.D. |
| 1b    | D2-rac | 40% | 20% |
| 1c    | D2(R)  | 25% | 40% |
| 1d    | D2(S)  | 70% | N.D. |
| 1e    | D3(R)  | 15% | 49% |
| 1f    | D3(S)  | 58% | N.D. |

Conditions: 1. TsOH (0.1 eq), DG, toluene, 80 °C;
2. Cu(OTf)₂ (1.3 eq), O₂, Na ascorbate
(2.0 eq), MeOH, acetone, 50 °C, 90 min;
3. Na₄EDTA, work up

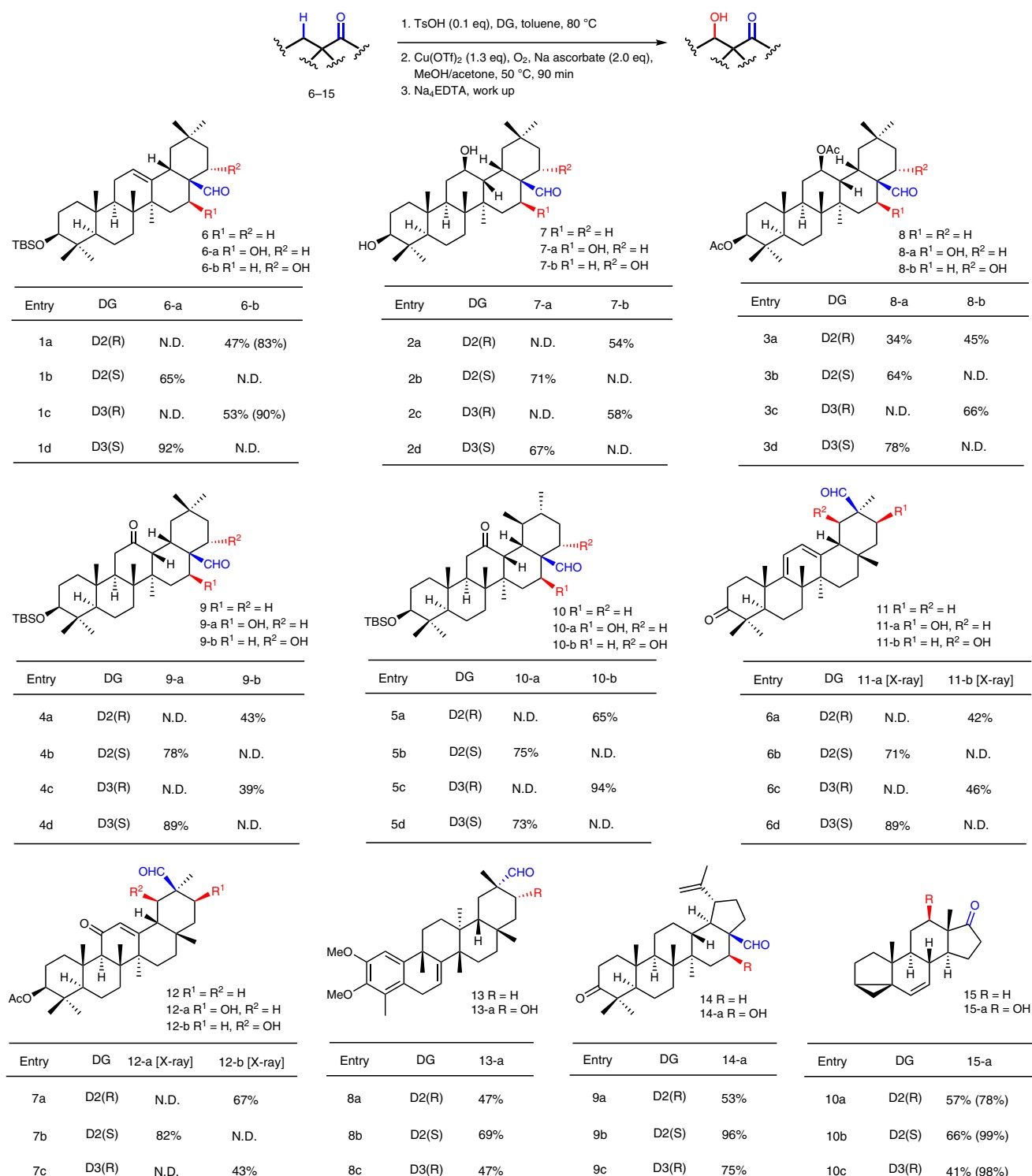

**Fig. 3 Scope of the site-selective C–H hydroxylation of PT derivatives.** The yield in parentheses are based on recovered starting materials. In entries 5c, 5d, and 9a-d, during working up, the mixture was treated with Na$_4$EDTA and followed with 1N HCl for 1h. Ac acetyl, N.D. not detected.

(**R**) as directing group, the site-selectivity of **5-b** was further enhanced, in that its yield was increased to 49% while the yield of **5-a** decreased to 15% (entry 1e). The use of the *S* enantiomer **D3** (*S*) reversed the selectivity, leading to **5-a** as the only C–H hydroxylated product (58%; entry 1f).

Next, we examined the substrate scope and functional group tolerance of this tunable transformation on PT derivatives

employing Baran's standard conditions (Fig. 3)[37], which involved imine formation (0.1 equiv. TsOH, toluene, 80 °C), Cu-mediated C–H hydroxylation (1.3 equiv. Cu(OTf)$_2$, 2.0 equiv. Na ascorbate, O$_2$, 50 °C), and removal of the imine auxiliary (work up with Na$_4$EDTA, or in some cases, followed with 1N HCl). Remarkably, hydroxylation of oleanane C28-aldehyde derivative **6**, which remains the innate C12–C13 double bond, was completely

site-selective, furnishing exclusively 22α-ol **6-b** (47%; 43% recovered **6**) in the presence of **D2(R)** or 16β-ol **6-a** (65%) in the presence of **D2(S)** (entries 1a and 1b). With the bulkier **D3(R)** and **D3(S)** as directing groups, the yield of **6-b** was slightly increased (53%), while the yield of **6-a** was dramatically increased to 92% (entries 1c and 1d). Similar results were obtained with oleanane C28-aldehydes **7–9** as the substrates, in which the C12–C13 double bond (in **6**) had been converted into C12-hydroxy, acetyloxy, and ketone, respectively. The site-selectivity was lost only when the C12-acetyloxy derivative **8** was used as substrate and **D2(R)** as directing group, leading to a mixture of **8-a** (34%) and **8-b** (45%; entry 3a); nevertheless, replacing **D2(R)** with **D3(R)** led to **8-b** (66%) exclusively (entry 3c). Ursane derivative **10** showed similar site-selectivity upon hydroxylation, nevertheless, in comparison to the reactions of its oleanane counterpart **9**, the yields of the 22α-ol product (**10-b**) were greatly increased (65% and 94%) in the presence of the *R*-configured directing groups (entries 5a and 5c).

Bearing a similar skeleton of oleananes but the aldehyde function at C30, glycyrrhetinic acid derivatives **11** and **12** were subjected to the present hydroxylation procedure. The *R*-configured directing groups, i.e., **D2(R)** and **D3(R)**, led to site-selective hydroxylation at C19, giving 19β-ol **11-b/12-b** exclusively (42~67%; entries 6a/7a and 6c/7c). The *S*-configured directing groups, i.e., **D2(S)** and **D3(S)**, resulted in hydroxylation only at C21, giving 21β-ol **11-a/12-a** in higher yields (62~89%; entries 6b/7b and 6d/7d).

Celastrol derivative **13**, which bears a disparate C/D ring conjunction and the aldehyde function at the α-oriented C29, was examined as a substrate. The hydroxylation took place only at C21, irrelevant to the chirality of the directing groups, providing 21α-ol **13-a** in good yields (47~80%; entries 8a–d). The chirality of the directing group indeed influenced the yields of **13-a**, with the *S*-configured directing groups being more favorable for the hydroxylation.

We next examined a lupane derivative, i.e., C28-aldehyde **14**, as a substrate, which has a *trans*-fused five-membered E ring. The hydroxylation took place only at C16 to provide **14-a**. Remarkably, **14-a** was obtained in nearly quantitative yields with the *S*-configured amines **D2(S)** and **D3(S)** as the directing groups (~97%; entries 9b and 9d), whereas it was obtained in lower yields with the *R*-configured amines **D2(R)** and **D3(R)** (53% and 75%; entries 9b and 9d).

Finally, we investigated steroidal Δ⁶-*i*-diene 17-ketone **15**, which was reported to be an inferior substrate for the hydroxylation[37]; under conditions here adopted, 12β-ol **15-a** was obtained in only 2% yield, or in 40% yield upon replacing Cu(OTf)$_2$ with Cu(MeCN)$_4$PF$_6$, with the achiral (pyridin-2-yl)methane-1-amine (**D1**) as directing group. Applying the present chiral amines in the reaction, compound **15** was converted into **15-a** cleanly, with the best isolated yield being 66% (34% recovered **15**) in the presence of **D2(S)** (entry 10b).

An eminent feature of the above transformations is the compatibility with various functional groups on the polycyclic skeletons, including hydroxyl, silyl ether, acetyl, olefin, ketone, and enone groups. Thus, a variety of PT intermediates can be readily prepared and utilized for further elaboration into complex natural PTs and their derivatives. Herein, three representative natural oleanane glycosides (**1–3**) bearing hydroxyl group at C16 or C22 and barringtogenol C (**4**) bearing hydroxyl groups at C16, C21, and C22 were successfully synthesized.

## Synthesis of representative PT saponins 1–3.

Oleanane 3-*O*-glycosides **1–3** are minor components from traditional Chinese medicinal plants, each contains an additional hydroxyl group on the

oleanane skeleton, i.e., the 16β-OH, 16α-OH, and 22α-OH, respectively. Saikosaponin E (**1**), showing antitumor and antiviral activities, is a biomarker of Chaihu (the roots of *Bupleurum* species), which is a commonly used traditional Chinese medicine[20,21]. Saponin **2** has been isolated from *Albizia inundata and A. anthelmintica*, which showed potent antitumor activities[22,23]. Kochianoside I (**3**) was isolated from the fruits of *Kochia scoparia*, which are mainly used as an antipruritogenic agent[24]. The synthesis of these natural PT glycosides commenced with scaling up the site-selective hydroxylation of **6**, which was readily prepared from oleanolic acid in two steps (86% yield) (Fig. 4). Without optimization of the previous conditions, the gram-scale hydroxylation of **6** in the presence of **D2(S)** gave 16β-ol **6-a** as the only isomer in 60% yield; the use of **D2(R)** provided 22α-ol **6-b** in 45% yield, again as the only isomer. The easy availability of **6-a** and **6-b** secured the subsequent transformations. Thus, reduction of aldehyde **6-a** with LiAlH$_4$ followed by NBS induced intramolecular etherification delivered the corresponding 12-bromide-13,28-ether, which underwent elimination under the action of DBU, providing olefin **16** (93% over three steps)[41]. Acetylation of the 16-OH followed by removal of the 3-*O*-silyl group gave the desired aglycone **17** (72%). Condensation of **17** with the readily available disaccharide imidate **18** (Supplementary Fig. 4) proceeded smoothly under the promotion of TBSOTf[42]; the resulting glycoside was then subjected to global removal of the acyl groups with KOH in MeOH, furnishing Saikosaponin E (**1**) in 73% yield over two steps (Supplementary Table 2).

Alternatively, aldehyde **6-a** was subjected to Pinnick oxidation and subsequent benzylation, providing ester **19** (84%). Exposure of **19**, which bears a β-OH at C16, to Dess-Martin periodinane gave the corresponding C16-ketone, which was subjected to reduction with NaBH$_4$ in ethanol to furnish the requisite 16α-ol **20** as the major product (74%), along with 16β-ol isomer **19** (14%)[43]. Masking the hydroxyl group as an acetyl ester followed by removal of the 3-*O*-silyl group gave rise to aglycone **21** (61%). Glycosylation of **21** with disaccharide imidate **22** (Supplementary Fig. 5) under the catalysis of triflic acid provided the coupled disaccharide **23** (81%)[44]. The *N*-phthaloyl group in **23** was dismantled and the resulting amine acetylated to give **24** (86% yield over two steps). Finally, successive hydrogenolysis of the benzyl ester and cleavage of the acetate (NaOMe, MeOH/THF) led to albizia saponin **2** (84% over two steps) (Supplementary Table 3).

Following similar transformations used in the above synthesis, aldehyde **6-b** was converted into ester **25** (81%) via Pinnick oxidation and subsequent benzylation. Acetylation of the 22α-OH and removal of the 3-*O*-silyl group in **25** gave the requisite aglycone **26** (79% over two steps). Condensation of **26** with thioglycoside **27** under the promotion of NIS and TMSOTf led to glycoside **28** (89%). The glucose residue in **28** was then transformed into a glucuronate unit in three steps (59%), i.e., regioselective opening of the 4,6-*O*-benzylidene group with BH$_3$/Cu(OTf)$_2$[45], oxidation of the resulting 6-OH with TEMPO/BAIB, and ester formation with BnBr. Finally, cleavage of the benzyl groups via hydrogenolysis over Pd/C followed by removal of the acyl groups with NaOMe in MeOH furnished kochianoside I (**3**) (79% over two steps) (Supplementary Table 4).

## Synthesis of barringtogenol C (4).

Barringtogenol C (**4**) was isolated from *Hydrocotyle ranunculoides*[25] and showed potent anti-inflammatory, anti-tumor, and anti-microbial activities[26]. Representing a typical PT bearing hydroxy groups at both D and E rings, barringtogenol C (**4**) became a target for examining the present methodology in sequential introduction of hydroxy

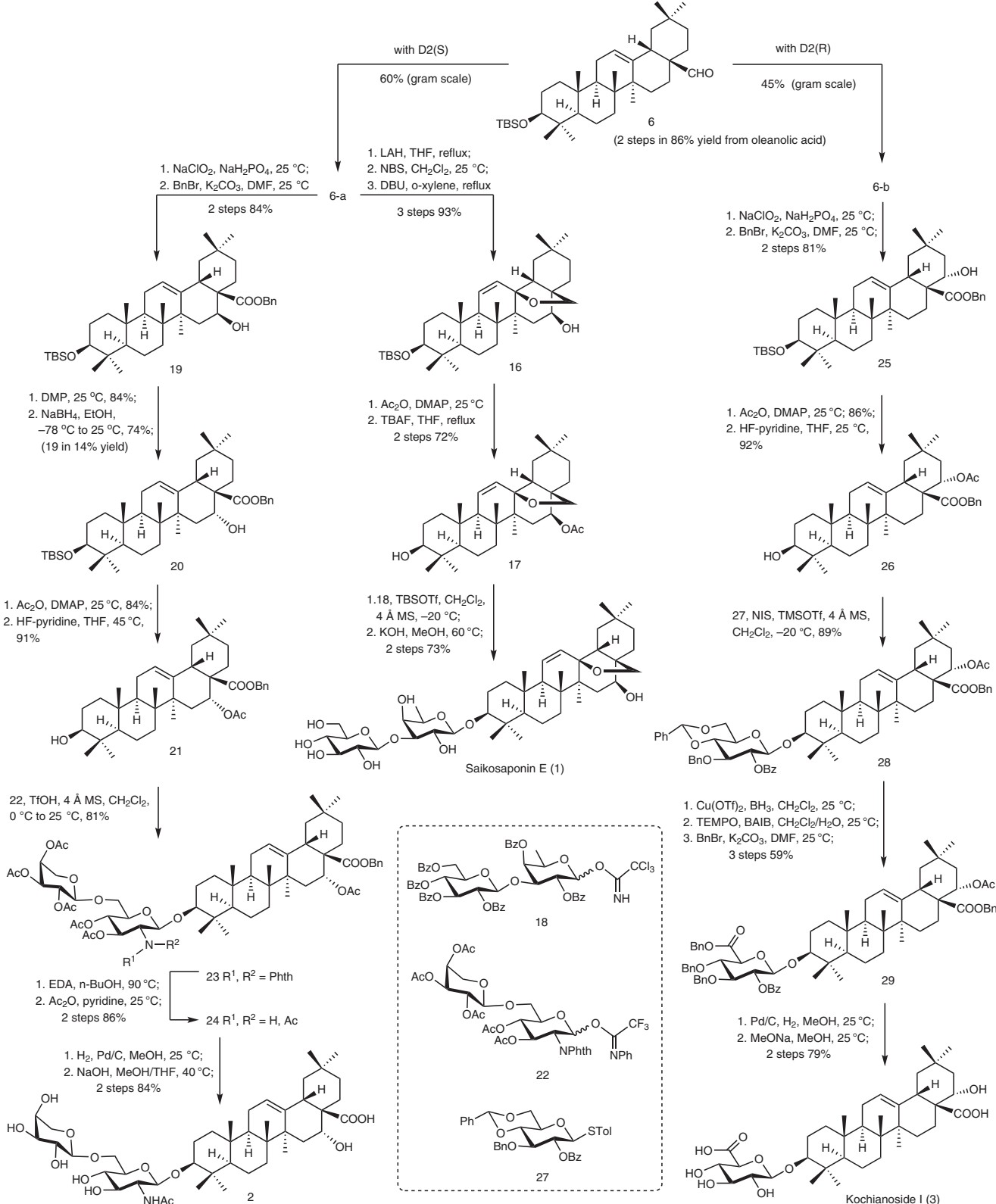

**Fig. 4 Synthetic applications.** Expeditious synthesis of Saikosaponin E (**1**), Albizia saponin **2**, and Kochianoside I (**3**). BAIB (diacetoxyiodo)benzene, Bn benzyl, DBU 1,8-diazabicyclo[5.4.0]undec-7-ene, DMAP 4-dimethylaminopyridine, DMF *N,N*-dimethylformamide, DMP Dess-Martin periodinane, EDA ethylenediamine, LAH lithium aluminum hydride, MS molecular sieves, NBS *N*-bromosuccinimide, NIS *N*-iodosuccinimide, Phth phthaloyl, TBAF tetrabutylammonium fluoride, TEMPO 2,2,6,6-tetramethylpiperidinooxy, THF tetrahydrofuran, TMS trimethylsilyl.

**Fig. 5 Synthetic applications.** Expeditious synthesis of barringtogenol C (**4**). TBHP *tert*-butyl hydroperoxide, Vo(acac)$_2$ vanadium-oxy acetylacetonate.

groups onto the D/E rings. Thus, the 22α-OH in **6-b**, which was introduced by the present hydroxylation in the presence of **D2** (**R**), was eliminated to give Δ$^{21,22}$ derivative **30** (72%) (Fig. 5). Compound **30** was subjected to reduction of the 28-aldehyde and subsequent epoxidation to provide β-epoxide **31** (87% over two steps). Then, the 28-hydroxyl group was converted to aldehyde to give **32** as a substrate for the second site-selective hydroxylation. Gratifyingly, subjection of **32** to the standard conditions with **D2** (**S**) as the directing group led to the desired 16β-ol **33** in a satisfactory 68% yield, with the epoxide remaining intact. The successful introduction of the 16-OH set a stage for further transformations into the natural product. Indeed, conversion of the 28-aldehyde into ester, inversion of the configuration of the 16β-OH, and acetylation of the resultant 16α-OH afforded **36** (75% over five steps). Under the action of BF$_3$·OEt$_2$ in aqueous toluene, the 21,22-β-epoxide in **36** was selectively opened to furnish 21β,22α-diol **37** (84%), through probably a *6-exo-trig* hydrolysis mediated by the 16α-*O*-acetyl group[46]. Finally, removal of the 3-*O*-TBS group and reduction of the 28-ester into alcohol and simultaneously cleavage of the 16-*O*-acetyl group furnished barringtogenol C (**4**) (92%) (Supplementary Tables 5 and 6).

## Discussion

The site-selective C–H hydroxylations at the D/E rings of PTs have been achieved by Schönecker and Baran's Cu-mediated aerobic oxidation procedure, with the site-selectivities being controlled by the chirality of the transient pyridine-imino directing group. Thus, introduction of the 16β-ol/22α-ol or 19β-ol/21β-ol onto the oleanane/ursane skeletons has become a convenient practice from the corresponding C28- or C30-aldehydes (i.e., **5**-**12**), which are readily accessible from the abundant feedstocks, such as oleanolic acid, ursanic acid, and glycyrrhetinic acid. The impact of the chirality of the directing group is also obvious in the hydroxylation of friedelane-type C29-

aldehyde (i.e., **13**) or lupane C28-aldehyde (i.e., **14**), where no regioselectivity is involved; the yields of the corresponding 21α-ol products (**13-a**) or 16β-ol (**14-a**) are significantly higher when the matched amine enantiomers are used. Besides the transient nature of the directing group[47,48], the excellent compatibility with various functional groups enables application of the present methodology to the preparation of advanced PTs derivatives which are amenable to further transformations. In this regard, three bioactive oleanane glycosides (**1**–**3**) from medicinal plants, which bear 16β-OH, 16α-OH, or 22α-OH, respectively, are conveniently synthesized from oleanane acid in a modicum of steps. Moreover, sequential hydroxylations, directed by a pair of the chiral amine, have been sucessfully performed, thus barringtogenol C (**4**) bearing characteristic 16,21,22-OHs is synthesized. In line with the already available methods for functionalization of the A/B rings of PTs[14–16,49], a way has now been paved toward the highly diverse natural PTs from abundant feedstocks, thus in-depth medicinal studies on PTs shall become feasible future projects.

## Methods

**General**. All reactions were carried out under argon with anhydrous solvents in flame-dried glassware. Reactions were monitored by thin layer chromatography (TLC) carried out on Millipore Sigma glass TLC plates (silica gel 60 coated with F254, 250 µm) using UV light for visualization and aqueous ammonium cerium nitrate/ammonium molybdate or basic aqueous potassium permanganate as developing agent. SiliaFlash® P60 silica gel (particle size 40–63 µm, pore size 60 Å) was used for flash column chromatography. NMR spectra were recorded on a Bruker Avance III 400 MHz or an Agilent DD2 500 MHz NMR spectrometer. IR spectra were recorded on a Thermo Scientific Nicolet 380 FT-IR spectrometer. Melting points are uncorrected and were recorded on an SGW X-4 apparatus. High-resolution mass spectra (HRMS) were recorded on a Bruker Apex III 7.0 Tesla FT-ICR, an IonSpec 4.7 Tesla FT-ICR, or a Waters Micromass GCT Premier mass spectrometer.

**Standard procedure for imine formation**. To a solution of the aldehyde or ketone substrate and *p*-toluenesulfonic acid monohydrate (0.10 equiv.) in toluene (0.10 M) in a flame-dried flask, was added amine **D2(S)** (3.0 equiv.). The mixture was heated

to 80 °C until imine formation was complete as monitored by $^1$H NMR (normally *ca.* 2 h for aldehyde and 10 h for ketone). The mixture was cooled to 25 °C and diluted with EtOAc (30 mL). The organic layer was washed sequentially with saturated aqueous $NH_4Cl$ (2 × 20 mL), saturated $NaHCO_3$ (1 × 20 mL), and brine (1 × 20 mL), and was then dried over anhydrous $Na_2SO_4$, filtered, and concentrated under vacuum. The crude product was used directly in the next step without further purification.

**Standard procedure for Schönecker–Baran oxidation.** The imine substrate (1.0 equiv.), copper(II) triflate (1.3 equiv.), and sodium L-ascorbate (2.0 equiv.) were added to a round-bottom flask. Acetone (0.05 M) and methanol (0.05 M) were added at 25 °C. The mixture was stirred for 5 min (the reaction mixture may turn brown). $O_2$ from a balloon was bubbled through the mixture for 5 min (resulting in a blue/green solution), and then the mixture was heated to 50 °C under an $O_2$ atmosphere for 1.5 h. The mixture was cooled to 25 °C, EtOAc (3 mL) and saturated aqueous $Na_4EDTA$ (6.0 mL, pH ~10) were added and the stirring continued for 0.5 h. The layers were separated. The aqueous layer was extracted with EtOAc (3 × 10 mL). The combined organic phase was washed sequentially with saturated $NaHCO_3$ (1 × 20 mL) and brine (1 × 20 mL), and was then dried over anhydrous $Na_2SO_4$, filtered, and concentrated under vacuum. The crude product was purified by flash column chromatography to give the hydroxylated product.

## Data availability

The authors declare that all data supporting the findings of this study are available within the paper and its supplementary information files, including experimental details, characterization data, and $^1$H and $^{13}$C NMR spectra of all new compounds (Supplementary Figs. 6–156). The X-ray crystallographic coordinates for structures **S20**, **S21**, **11-a**, **11-b**, **12-a**, and **12-b** have been deposited at the Cambridge Crystallographic Data Centre (CCDC), under deposition numbers CCDC1868411, CCDC1885648, CCDC1868416, CCDC1868439, CCDC1885629, and CCDC1885628, respectively. These data can be obtained free of charge from The Cambridge Crystallographic Data Centre via www.ccdc.cam.ac.uk/data_request/cif.

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

## Acknowledgements
This work is financially supported by the National Key Research & Development Program of China (2018YFA0507602), the National Natural Science Foundation of China (21621002 and 21871290), the Key Research Program of Frontier Sciences of the Chinese Academy of Sciences (ZDBS-LY-SLH030), and the Strategic Priority Research Program of the Chinese Academy of Sciences (XDB20020000).

## Author contributions
B.Y. and T.M. conceived the work. T.M. and B.W. conducted the synthetic work. T.M., B.W., and D.Z. conducted the data analysis. B.Y. and T.M. wrote the manuscript.

## Competing interests
The authors declare no competing interests.
