## [Peer Review File · Nature Communications]

REVIEWER COMMENTS

Reviewer #1 (Remarks to the Author):

Yu and co-workers report on the regioselective C-H hydroxylation of steroid-like skeletons. The Cu-mediated oxidation is based on seminal work by Schönecker and employs chiral pyridine-methylamines as directing groups. Directing groups are omnipresent in C-H oxidation reactions and I do not see the innovative aspect of the presented work in this part. However, it is interesting to note that previous work by Baran and Garcia-Bosch did not investigate the role of chiral versions of these ligands to control the regio-selectivity. The observed regio-selectivities/obtained yields are generally good as shown for a panel of substrates. The application of the presented method enabled the authors to access barringtogenol C, kochianoside I, saikosaponin and albizia spanonin from oleanolic acid in a few steps. It might be interesting to the reader where large quantities of the starting material can be obtained (up to 100g of oleanolic acid are used in the SI). Only small quantities (500 mg) are available from most standard suppliers (exception: carbosynth)! The presented approaches clearly benefit from brevity to reach the target compounds. However, the chiral pool strategy might prevent access to analogs carrying deep-seated structural modifications. In contrast to transient-strategies (Yu, Science 2016), two steps are required for the installation and removal of the directing group. The scholarly presentation of the Schemes is of good quality, however, some further improvement and optimization is recommended for certain parts. In addition, the manuscript needs thorough language polishing and corrections as mentioned below.

Corrections and Comments:

Abstract: replace "family" with "families"

Abstract: rewrite "site-selective on PT skeletons as tuned unexpectedly"

Figure 2: replace "iPr" with "i-Pr"

Do not use "RT" or "rt" but specify exact temperatures throughout the manuscript,

Figure 5, third reaction: replace "DMP." with "DMP, "

Figure 5, last reaction replace "50°C" with "50 °C"

Page 7: replace "(75% over 5 steps)." with "(75% over five steps)."

Page 2: remove "cleanly"

Page 3: insert: oxygen into the conditions in brackets for Cu-mediated C-H hydroxylation

Page 3: replace "recovery yield" with the percentage of recovered starting material

Figure 3: confusing: Entries and molecules have partially same numbers, should be avoided

Page 4: add "D1" to "(pyridin-2-yl)methane-1-amine"

Supporting Information:

Equivalents are missing for reagents and reactants

A reaction (mixture) can't be quenched, but only excess reagents or reactants. Please correct!

Reviewer #2 (Remarks to the Author):

The authors synthesized triterpenoid natural products utilizing C-H oxidation methods. This reviewer thinks that this manuscript is not sufficient for publication in JACS or Angewandte but admittedly I also don't read this Journal and don't know more than authors pay large sums of money to publish here. Thus the suitability for publication here is at the Editor's discretion. All employed chemical methods and strategies are well established, no fundamental discovery and scientific advancement were made. The synthesized natural products have little academic and industrial significance. This reviewer also requests the following items to improve the quality of

this manuscript:

Figure 1A there are two solid and dashed arrows from 28 to 26 and 22, meaning the corresponding directed oxidations are known and unknown. Please correct the figure so that this is easier to be interpreted by the readership.

"Thus, further elaboration of the hydroxylated products becomes a convenient task." This sentence is missing a connection from the previous sentence. Please rephrase.

I appreciate the authors put all failed attempts in SI, which would be useful resources for the readership. I would like to ask for preparing summary figures for failed attempts in SI, rather than experimental style page by page schemes. This would be more easy to understand and see the whole picture of strategies and attempts.

Reviewer #3 (Remarks to the Author):

This communication combines my interest in CH oxidation (1) of easily-available stereodefined terpenoidal natural products (2) using a synthetic strategy that provides a value-added product (increase in intricacy of the oxygen heteroatom and a new stereocenter (3). In addition, it synthesizes a seminal collection of biologically active targets using aldimines prepared from heterobenzylic pyridyl amines as bidentate ligands for copper mediated chemspecific oxidation. The finding that enantiopure amines effect double-stereoselective regiocontrol makes this paper a must-read for advanced students studying synthetic chemistry in the 21st century. The manuscript is well-organized, clearly presented, and fully-supported in the experimental material. My congratulations for an outstanding contribution.

- (1) Handbook of Reagents for Organic Synthesis: Reagents for Direct Functionalization of C-H Bonds Ed. P. Fuchs, John Wiley & Sons, 2007, 424 pp.
- (2) Chemistry of Trisdecacyclic Pyrazine Antineoplastics: The Cephalostatins and Ritterazines Seongmin Lee, Thomas G. LaCour, and Philip L. Fuchs, Chem Rev. **2009**, 2275–2314. doi: 10.1021/cr800365m
- (3) Increase in Intricacy - A Tool for Evaluating Organic Syntheses Philip L.Fuchs Tetrahedron **57**, 2001, 6855-6875 DOI 10.1016/S0040-4020(01)00474-4

Adding a mechanistic scheme showing the Baran oxidation (especially showing a double stereoselective transition state) would greatly improve the reader's ability to understand this work.

If available, a comment about potential failure modes in reactions of enolizable aldehydes (beta-hydroxy aldimine beta-elimination/hydrolysis/ 1,4 addition etc. or aldol rxn?) would be highly instructive. In addition, your *model studies with 5-12 prompt the question as to whether a second hydroxyl can be subsequently added to the steroidal skeleton using the complementary aldimine.*

Annotations of English errors are highlighted in yellow and suggested deletions indicated in red on a copy of the manuscript.

Philip L. Fuchs
R. B. Wetherill Professor of Chemistry, Emeritus
Purdue University
West Lafayette, IN 47906

I. Response to Decision Letter (NCOMMS-20-18870)

We are grateful to Prof. Fuchs and the other two reviewers for their kind comments and helpful suggestions that have helped us to improve our paper. As indicated in the point-by-point responses that follow, we have taken all these comments into account in the revised manuscript. The response to the comments of Prof. Fuchs could be published if allowed.

Reviewer #1 (Remarks to the Author):

Yu and co-workers report on the regioselective C-H hydroxylation of steroid-like skeletons. The Cu-mediated oxidation is based on seminal work by Schönecker and employs chiral pyridine-methylamines as directing groups. Directing groups are omnipresent in C-H oxidation reactions and I do not see the innovative aspect of the presented work in this part. However, it is interesting to note that previous work by Baran and Garcia-Bosch did not investigate the role of chiral versions of these ligands to control the regio-selectivity. The observed regio-selectivities/obtained yields are generally good as shown for a panel of substrates. The application of the presented method enabled the authors to access barringtonol C, kochianoside I, saikosaponin and albizia spanonin from oleanolic acid in a few steps. It might be interesting to the reader where large quantities of the starting material can be obtained (up to 100 g of oleanolic acid are used in the SI). Only small quantities (500 mg) are available from most standard suppliers (exception: carbosynth)! The presented approaches clearly benefit from brevity to reach the target compounds. However, the chiral pool strategy might prevent access to analogs carrying deep-seated structural modifications. In contrast to transient-strategies (Yu, Science 2016), two steps are required for the installation and removal of the directing group.

The scholarly presentation of the Schemes is of good quality, however, some further improvement and optimization is recommended for certain parts. In addition, the manuscript needs thorough language polishing and corrections as mentioned below.

Response: Thanks. The suggested corrections as mentioned below have been corrected.

Oleanolic acid is the most abundant triterpene in Nature, it can be purchased in large quantities at very low prices from the producers. It can also be purchased from general chemical suppliers at higher prices, *e. g.*, 100g/~70\$ from Chinese supplier Macklin (<http://www.macklin.cn/products/O815225>), 100g/106\$ from American supplier Accela (<http://www.accelachem.com/cn/search.html?keyword=508-02-1&attname=0>), and 25g/215\$ from Japanese supplier TCI (<https://www.tcichemicals.com/US/en/search/?text=508-02-1>).

For the transient-strategies, the early reference by Yu et al. (Science 2016) has been added as ref. 50. Indeed, the imine directing group here is operationally but not mechanistically transient (ref. 49); it is installed, used in the C-H hydroxylation, and removed in one pot.

Corrections and Comments:

Abstract: replace “family” with “families”

Response: This has been corrected.

Abstract: rewrite “site-selective on PT skeletons as tuned unexpectedly

Response: This sentence has been rephrased, it now reads: We find that Schönecker and Baran’s Cu-mediated aerobic oxidation can be applied and become site-selective on PT skeletons, as being effected unexpectedly by the chirality of the transient pyridine-imino directing groups.

Figure 2: replace “iPr” with “i-Pr”

Response: This has been revised.

Do not use “RT” or “rt” but specify exact temperatures throughout the manuscript,

Response: The exact temperatures have been provided instead of “RT” and “rt” throughout the manuscript.

Figure 5, third reaction: replace “DMP.” with “DMP, “

Response: This typo in Fig. 5 has been corrected.

Figure 5, last reaction replace “50°C” with “50 °C”

Response: This has been revised.

Page 7: replace “(75% over 5 steps).” with “(75% over five steps).”

Response: This has been revised.

Page 2: remove “cleanly”

Response: This has been revised.

Page 3: insert: oxygen into the conditions in brackets for Cu-mediated C-H hydroxylation

Response: The oxygen has been inserted.

Page 3: replace “recovery yield” with the percentage of recovered starting material

Response: The recovery yield has been replaced with the percentage of the recovered starting material.

Figure 3: confusing: Entries and molecules have partially same numbers, should be avoided

Response: The compound numbers have been changed from “**1a, 1b, 2a, 2b**, etc.” to “**1-a, 1-b, 2-a, 2-b**, etc.” In addition, the compound numbers are in bold.

Page 4: add ”D1” to “(pyridin-2-yl)methane-1-amine”

Response: Yes, “D1” has been added.

Supporting Information:

Equivalents are missing for reagents and reactants

Response: We have added the equivalents for reagents and reactants.

A reaction (mixture) can’t be quenched, but only excess reagents or reactants. Please correct!

Response: We have done the correction accordingly.

Reviewer #2 (Remarks to the Author):

The authors synthesized triterpenoid natural products utilizing C–H oxidation methods. This reviewer thinks that this manuscript is not sufficient for publication in JACS or Angewandte but admittedly I also don't read this Journal and don't know more than authors pay large sums of money to publish here. Thus the suitability for publication here is at the Editor's discretion. All employed chemical methods and strategies are well established, no fundamental discovery and scientific advancement were made. The synthesized natural products have little academic and industrial significance. This reviewer also requests the following items to improve the quality of this manuscript:

Response: This work reports the following new findings, which we think are significant: 1) The extension of Schönecker-Baran's Cu-mediated aerobic oxidation to the hydroxylation of pentacyclic triterpenoids. This is significant, because the previous application of the Schönecker-Baran method is limited to providing steroids bearing C12 oxidations (ref. 35-39). 2) The site-selective hydroxylation effected by the chirality of the directing group. For the first time, it is find that the chirality of the directing group can tune the site-selectivity of the Schönecker-Baran oxidation. Moreover, a practically useful scope has been demonstrated. 3) The first synthesis of barringtogenol C, kochianoside I, saikosaponin E, and albizia saponin based on the methodological advancement. These compounds represent typical structures of the natural pentacyclic triterpenoids and are reported to have potent biological activities (ref. 20-26). In fact, pentacyclic triterpenoids constitute the largest family of natural products and have attracted a great attention of medicinal and natural product scientists (ref. 1-6). Several of these natural products have been widely used in clinic, such as glycyrrhizic acid, many are under development at different stages (ref. 1), such as the well-known immune-adjuvant QS21. It is true that less attention has been paid to these compounds by synthetic chemists. The lack of synthetic advances has hampered the medicinal and biological studies on pentacyclic triterpenoids. This point further proves the significance of the present work.

Figure 1A there are two solid and dashed arrows from 28 to 26 and 22, meaning the corresponding directed oxidations are known and unknown. Please correct the figure so that this is easier to be interpreted by the readership.

Response: Thanks for the suggestion. We have deleted the solid arrows from C28 to C26 and C22 in oleananes in Fig. 1A. These two arrows were originally drawn to indicate previous transformations to introduce C15-16 and C21-22 double bonds via palladium promoted dehydrogenation (ref. 18).

"Thus, further elaboration of the hydroxylated products becomes a convenient task." This sentence is missing a connection from the previous sentence. Please rephrase.

Response: Yes, we have rephrased the sentences, it now reads: An eminent feature of the above transformations is the compatibility with various functional groups on the polycyclic skeletons, including hydroxyl, silyl ether, acetyl, olefin, ketone, and enone groups. Thus, a variety of PT intermediates can be readily prepared and utilized for further elaboration into complex natural PTs and their derivatives.

I appreciate the authors put all failed attempts in SI, which would be useful resources for the readership. I would like to ask for preparing summary figures for failed attempts in SI, rather than experimental style page by page schemes. This would be more easy to understand and see the whole picture of strategies and attempts.

Response: Yes, we have added a summary figure (Fig. S2) for the failed attempts in SI.

Reviewer #3 (Remarks to the Author):

This communication combines my interest in CH oxidation (1) of easily-available stereodefined terpenoidal natural products (2) using a synthetic strategy that provides a value-added product (increase in intricacy of the oxygen heteroatom and a new stereocenter (3)). In addition, it synthesizes a seminal collection of biologically active targets using aldimines prepared from heterobenzylic pyridyl amines as bidentate ligands for copper mediated chemspecific oxidation. The finding that enantiopure amines effect double-stereoselective regiocontrol makes this paper a must-read for advanced students studying synthetic chemistry in the 21st century. The manuscript is well-organized, clearly presented, and fully-supported in the experimental material. My congratulations for an outstanding contribution.

(1) Handbook of Reagents for Organic Synthesis: Reagents for Direct Functionalization of C-H Bonds Ed. P. Fuchs, John Wiley & Sons, 2007, 424 pp.

(2) Chemistry of Trisdecacyclic Pyrazine Antineoplastics: The Cephalostatins and Ritterazines Seongmin Lee, Thomas G. LaCour, and Philip L. Fuchs, *Chem Rev.* 2009, 2275–2314. doi: 10.1021/cr800365m

(3) Increase in Intricacy - A Tool for Evaluating Organic Syntheses Philip L. Fuchs *Tetrahedron* 57, 2001, 6855-6875 DOI10.1016/S0040-4020(01)00474-4

Response: We are grateful to Prof. Fuchs for his kind comments.

Adding a mechanistic scheme showing the Baran oxidation (especially showing a double stereoselective transition state) would greatly improve the reader's ability to understand this work.

Response: Garcia-Bosch, Baran, and co-workers have proposed a mechanism of this reaction based extensive experimental evidences, including spectroscopic characterization, kinetic analysis, intermolecular reactivity, and radical trapping. The key elementary steps involve a mononuclear LCuII(OOR) intermediate which undergoes homolytic O–O cleavage to generate reactive RO• species, and the resultant RO• species are responsible for the C–H hydroxylation within the solvent cage (ref. 39, Fig. RL_1A). Based on this mechanism, we depict the key transition states which might determine the site-selectivity in Fig. RL_1B & C.

A) Key transition states proposed by Garcia-Bosch, Baran, and co-workers

B) The logic of site-selectivity

C) The plausible transition states responsible for the present site-selectivity

Fig. RL-1

While the logic for the site-selectivity is simple, which is determined by the biased orientation of the chiral directing group installed on the chiral substrate (Fig. RL_1B), the real transition state determining the site-selective C-H hydroxylation is not known. We have subjected these possible transition states to DFT calculation in the lab of a renowned expert, unfortunately, supportive results are not obtained. In addition, we failed in acquiring any reaction intermediate in the reaction. The lack of evidence make us unconfident to provide a mechanistic scheme in the article.

If available, a comment about potential failure modes in reactions of enolizable aldehydes (beta-hydroxy aldimine beta-elimination/hydrolysis/1,4 addition etc. or aldol rxn?) would be highly instructive. In addition, your model studies with 5-12 prompt the question as to whether a second hydroxyl can be subsequently added to the steroidal skeleton using the complementary aldimine.

Response: These are two interesting and important questions. The enolizable aldehydes, which might undergo the side-reactions as mentioned by Prof. Fuchs, have not been examined in the present study, nor have been reported previously (ref. 35-40). The substrate scope of Schönecker-Baran's Cu-mediated aerobic C-H hydroxylation is previously limited to steroidal 17-ketones (e.g., **15**) and camphor, and is now extended to PTs bearing an (not enolizable) angular aldehyde.

At the stage of methodological study, we did have tried sequential hydroxylations. Unfortunately, the second hydroxyl group could not be introduced in the presence of the first (*i.e.*, **6-b**→**C** and **5-a**→**F**; Fig. RL-2), and the starting hydroxy aldehyde was largely recovered. When the hydroxyl group was masked with mesyl ester, the second glycosylation took place smoothly (*i.e.*, **A**→**B** and **D**→**E**).

Fig. RL-2

In the synthesis of barringtonol **C** (**4**), the 22 α -OH is firstly introduced onto PT aldehyde **6** with **D2(R)** as the directing group; with **D2(S)** as the directing group, the 16 β -OH

is successfully introduced onto 21,22-epoxy-28-aldehyde **32** (Fig. RL_3).

Fig. RL-3

Annotations of English errors are highlighted in yellow and suggested deletions indicated in red on a copy of the manuscript.

Response: We are grateful to Prof. Fuchs for his kind suggestions. The suggested deletions in red have been deleted.

II. Other revisions

- 1) A “Data Availability” section has been added before the References.

REVIEWERS' COMMENTS:

Reviewer #3 (Remarks to the Author):

The corrected text answers my questions and should be published along with my comments as the authors request.